**Data Availability Statement:** The data underlying the results presented in the study are available

# Asthma in a prospective cohort of rural pregnant women from Sri Lanka: Need for better care during the pre-conceptional and antenatal period

Shashanka Rajapakse[1], Nuwan Wickramasinghe[2], Janith Warnasekara[2], Parami Abeyrathna[3], Gayani Amarasinghe[2], Ayesh Umeshana Hettiarachchi[2], Imasha Upulini Jayasinghe[2], Iresha Koralegedara[4], Thilini Chanchala Agampodi[2], Suneth B. Agampodi[2,5]*

1 Faculty of Medicine and Allied Sciences, Department of Physiology, Rajarata University of Sri Lanka, Mihintale, Sri Lanka, 2 Faculty of Medicine and Allied Sciences, Department of Community Medicine, Rajarata University of Sri Lanka, Mihintale, Sri Lanka, 3 Faculty of Medicine and Allied Sciences, Department of Family Medicine, Rajarata University of Sri Lanka, Mihintale, Sri Lanka, 4 Faculty of Medicine and Allied Sciences, Department of Anatomy, Rajarata University of Sri Lanka, Mihintale, Sri Lanka, 5 Department of Internal Medicine, Section of Infectious Diseases, School of Medicine, Yale University, New Haven, CT, United States of America

* suneth.agampodi@yale.edu

## Abstract

### Objectives

To describe the epidemiology and the effect of asthma on pregnancy outcomes in pregnant women from a rural geography.

### Methods

We conducted a prospective cohort study in Anuradhapura district, Sri Lanka enrolling all eligible pregnant women registered in the maternal care program. An interviewer-administered questionnaire-based symptom analysis and clinical assessment was conducted in the first and second trimesters.

### Results

We recruited 3374 pregnant women aged 15–48 years at conception. Self-reported physician-diagnosed asthma prevalence was 6.6% (n = 223) with only 41.7% (n = 93) on regular medical follow-up for asthma. The prevalence of wheeze reduced from pre-pregnancy (67.0%) to the first (46.4%) and second trimesters (47.7%; p<0.01). Of the 73 asthmatic women who did not have wheeze in the last 3 months preceding pregnancy, new-onset wheeze was reported by 6(8.2%) and 12(16.4%) in the first and second trimester, respectively. Pregnant women who sought medical care for asthma in the private sector had a lower likelihood of developing new-onset wheeze in the first trimester (p = 0.03; unadjusted OR = 0.94;95%CI 0.89–0.99). Thirty-four (33.3%) pregnant women had at least one hospital

under the doi: 10.5281/zenodo.5792499 from the zenedo repository at https://zenodo.org/record/5792499#.YcAeAGhBxhE.

**Funding:** Authors SA and TA received funding for this research project. This research was supported by the Accelerating Higher Education Expansion and Development (AHEAD) Operation of the Ministry of Higher Education, Sri Lanka funded by the World Bank. The funding agency had no role in the design of the study and collection, analysis, and interpretation of data and in writing the manuscript. The authors wish to acknowledge the funding agency (grant identifier: DOR STEM HEMS [6026-LK/8743-LK]).

**Competing interests:** The authors have declared that no competing interest exist.

admission due to exacerbation of wheeze during the first and second trimester. The prevalence of low birth weight (16.0%) was higher among pregnant asthmatic women.

## Conclusion

This study reports the high prevalence of asthma and asthma-associated pregnancy outcomes in women from a rural geography signifying the importance of targeted management.

## Introduction

Asthma is the most common chronic disease complicating pregnancy [1]. It commonly manifests as frequent episodes of wheezing, dyspnoea and nocturnal cough. Anatomical and physiological changes of pregnancy such as the increases in uterine size, intra-abdominal pressure and the subcostal angle results in reduced total lung volume and functional residual capacity [2]. Furthermore, hormonal changes of pregnancy result in increased sensitivity of the respiratory centre to carbon dioxide, increased minute ventilation and tidal volume [3]. In addition, changes in immunological functions, nutritional factors and female fetus are known to affect the control of asthma in pregnant women [4]. In vitro studies have demonstrated increased production of pro-inflammatory cytokines and activation of inflammatory pathways in pregnancy that may contribute to exacerbation of asthma [5]. Furthermore, significant differences in the plasma protein patterns of asthmatic and non-asthmatic pregnant women and increased levels of circulating immune cells have been demonstrated [6,7]. Although, the exact mechanism is unknown, these changes are implicated in worsening of asthma symptoms during pregnancy [8].

Poor asthma control and frequent asthma exacerbations during pregnancy are associated with adverse foetal, maternal and pregnancy outcomes such as low birth-weight, preterm birth, congenital malformations, and increased perinatal mortality [9]. Importantly, current evidence suggests that well-controlled asthma during pregnancy reduces the risk of adverse pregnancy outcomes. If asthma symptoms are adequately controlled, babies born to asthmatic mothers may achieve growth and development similar to that of babies born to non-asthmatic mothers [10]. Current evidence supports patient education, objective asthma control assessment, regular follow-up and active asthma management during pregnancy to improve asthma symptom control and to minimize adverse pregnancy outcomes [11].

However, adequate information on the prevalence of asthma among pregnant women from rural geographic regions of low and middle-income countries, control of asthma during pregnancy and the effect of asthma control on pregnancy outcomes is severely lacking. Asthma prevalence significantly varies according to the geography with a higher prevalence of severe asthma in low-income countries [12]. Against this background, this study aims to assess the epidemiology of asthma in pregnancy and the effect of asthma control on pregnancy outcome in a diverse cohort of pregnant women from geographically the largest district in Sri Lanka, which has one of the best maternal healthcare programmes in the region.

## Methods

This study was conducted as part of a large prospective cohort study of pregnant women: Rajarata Pregnancy Cohort (RaPCo) [13]. All eligible pregnant women registered with the

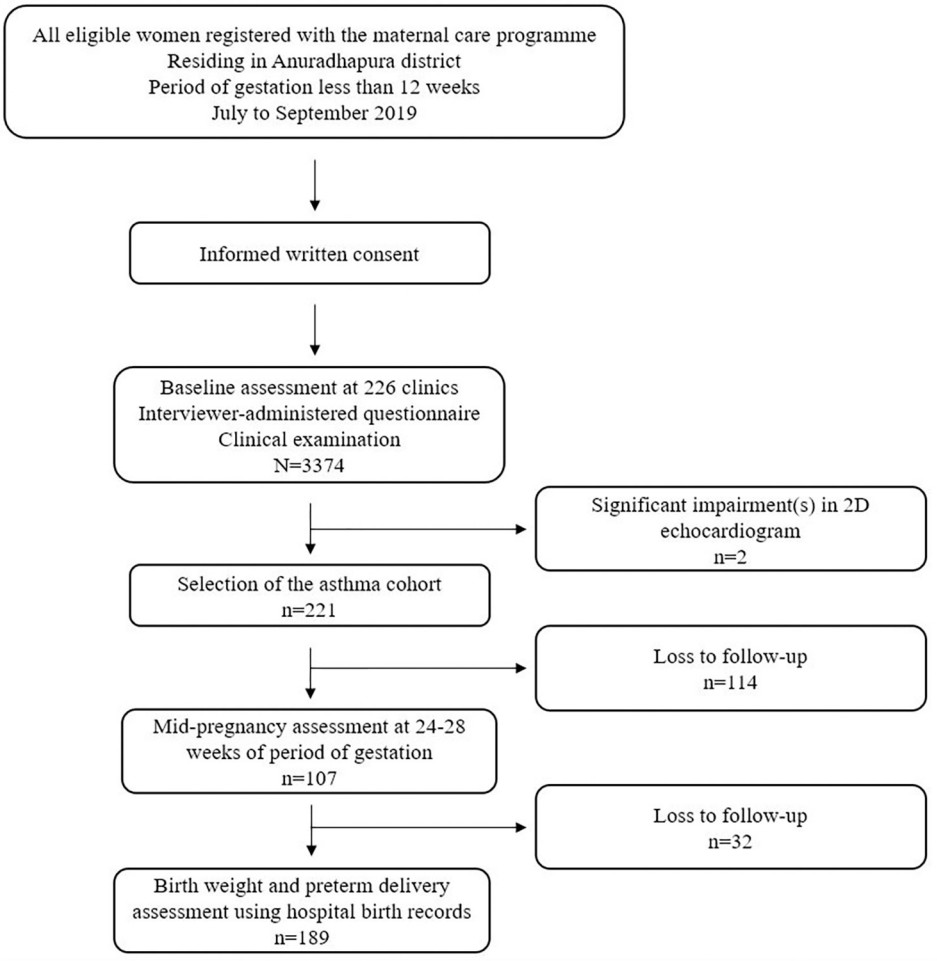

**Fig 1. Flow chart of the research.**

maternal care program (almost 100% of pregnant mothers are registered with the routine program) and were residing in Anuradhapura district and having a period of gestation less than 12 weeks were invited to the study (Fig 1). The study cohort included pregnant women presented to the program from July to September 2019. Baseline assessment was done in 226 special clinics conducted over 71 days for this purpose, which was coupled with the routine investigation process to maximize participation. Informed written consent was obtained before recruiting participants for the study. Assent and informed consent from the accompanying parent/ guardian were obtained for the research participants who are aged less than 18 years. Medically trained data collectors completed interviewer-based questionnaires to collect data on symptomatology and conducted clinical examination in all research participants. Data collection was conducted in the first and second trimester for each participant. All recruited pregnant women were invited to visit the clinic around 24–28 weeks for the mid-pregnancy assessment. Participants (n = 2) with comorbid significant 2D echo-cardiogram findings (pulmonary hypertension and ostium secundum atrial septal defect with right to left shunt) were excluded from the analysis to eliminate the role of cardiac dysfunction on the symptoms.

An interviewer-administered questionnaire to assess demographic factors, physical health status, symptoms of asthma and treatment seeking behaviour was used in this study after

obtaining informed written consent. A positive response to the question "Have you ever been diagnosed and confirmed as having asthma?" was considered as self-reported physician-diagnosed asthma (hereafter referred to as asthma). Asthma status in participants without documented evidence was confirmed following probing for a history of inhaler use, nebulization, exacerbating factors and hospitalization. A positive response to the question "Have you experienced shortness of breath in the last three months?" was considered dyspnoea and a positive response to the question "Was the shortness of breath increased with exertion?" is considered exertional dyspnoea. A positive response to the question "Have you experienced wheezing in the last three months?" was considered wheeze and a positive response to the question "Was the wheezing increased with exertion in the last three months?" was considered exertional wheeze. New-onset of symptoms was defined as an asthmatic participant who did not have a particular symptom during the last three months preceding pregnancy and developing the symptom during the first or second trimester.

Data analysis was conducted using SPSS beta version (IBM, Armonk, NY, USA) statistical software. Univariate analysis was performed to analyse the effect of housing conditions and domestic fuel source on pre-existing asthma in pregnant women. Bivariate analysis was performed to assess significant associations of asthma control during pregnancy with socio-demographic factors, patient characteristics, exposure to allergens, risk factors for asthma, health-seeking behaviour and patient education. Chi-square test for proportions was used to compare the estimates during pre-pregnancy, first trimester and second trimester with a significant level of 0.05. Odds ratio was calculated with 95% confidence interval to quantify the associations between symptoms of asthma and for pregnancy outcomes (low birth weight and preterm delivery). In addition, for pregnancy outcomes, the relative risk was computed with 95% confidence interval.

Ethical approval for the study was granted by the Ethics Review Committee of Faculty of Medicine and Allied Sciences, Rajarata University of Sri Lanka (ERC/ 2019/ 07).

## Results

We collected data from 3374 pregnant women aged 15 to 48 years at conception. Of the 3374 participants, 221 was selected as the asthmatic cohort (2 research participants with significant findings on 2D echocardiogram were excluded). Out of the 221 research participants with asthma, mid-pregnancy assessment at 24–28 weeks is available for 107 participants. The pregnancy outcomes—birthweight and preterm delivery—were assessed using hospital birth records. Table 1 shows the profile of study participants. The prevalence of asthma was 6.6% (n = 223; 95%CI 5.8%–7.5%) with 22.9% (n = 51) participants having documented evidence of a diagnosis of asthma (Table 2). The majority sought medical care for asthma in the private sector (n = 86; 46.7%), followed by the government sector (n = 84; 45.7%). Only 41.7% (n = 93) of the asthmatic pregnant women reported regular medical follow-up for asthma. Of the 223 pregnancies, 28.3% (n = 63) were unplanned with 2.3% (n = 5) due to contraceptive failure.

The prevalence of non-cardiac origin dyspnoea increased from the last three months before pregnancy 20.4% (n = 45; 95%CI 15.3%–26.3%) to the first (n = 51, 23.3%) and second trimester (n = 30, 28.3%) ($p > 0.11$) (Table 3). The total number of patients complaining of daily dyspnoea increased from pre-pregnancy (n = 8, 19.5%) to the first trimester (n = 17, 35.4%, $p = 0.10$). None of the study participants with asthma reported a frequency of dyspnoea symptoms more than once a week in the second trimester (Fig 2). In most study participants with asthma who had dyspnoea in the last three months preceding pregnancy (n = 30, 93.8%), the

**Table 1.** Demographic characteristics and housing conditions of asthmatic pregnant women without comorbid cardiac conditions (n = 221) and non-asthmatic pregnant women recruited to the study (N = 3151).

| Characteristic | Asthmatic | | Non-asthmatic | |
|---|---|---|---|---|
| | Number | Prevalence (%) | Number | Prevalence (%) |
| **Ethnicity** | | | | |
| Sinhalese | 205 | 92.8 | 2731 | 86.7 |
| Moor | 12 | 5.4 | 369 | 11.7 |
| Other | 4 | 1.8 | 51 | 1.6 |
| **Religion** | | | | |
| Buddhism | 199 | 90.0 | 2704 | 85.8 |
| Islam | 14 | 6.3 | 388 | 12.3 |
| Other | 8 | 3.6 | 59 | 1.9 |
| **Highest education level at school** | | | | |
| Up to or less than grade 10 | 26 | 11.7 | 351 | 11.2 |
| Grade 11 | 99 | 44.8 | 1548 | 49.4 |
| Grade 12–13 | 96 | 43.5 | 1232 | 39.3 |
| **Post-school educational qualifications** | | | | |
| Certificate course | 57 | 26.3 | 675 | 21.7 |
| Degree | 26 | 12.0 | 295 | 9.5 |
| **Age at conception (years)** | | | | |
| Less than 20 | 14 | 6.3 | 240 | 7.6 |
| 20–24 | 39 | 17.7 | 645 | 20.5 |
| 25–29 | 72 | 32.6 | 1096 | 34.8 |
| 30–34 | 67 | 30.3 | 751 | 23.8 |
| 35–39 | 25 | 11.3 | 350 | 11.1 |
| 40–44 | 4 | 1.9 | 66 | 2.1 |
| More than 44 | 0 | 0 | 3 | 0.1 |
| **Number of bedrooms in the house** | | | | |
| One | 19 | 8.6 | 244 | 7.8 |
| Two | 56 | 25.5 | 698 | 22.4 |
| Three | 83 | 37.7 | 1137 | 36.5 |
| Four or more | 62 | 28.3 | 1036 | 33.3 |
| **Number of other people sharing the bedroom** | | | | |
| One | 86 | 40.0 | 1184 | 39.2 |
| Two | 75 | 34.9 | 1257 | 41.7 |
| Three or more | 54 | 25.1 | 576 | 19.1 |
| **Major fuel source for cooking** | | | | |
| Firewood | 109 | 49.5 | 1549 | 49.7 |
| Liquid petroleum gas | 107 | 48.6 | 1480 | 47.5 |
| Other | 4 | 1.9 | 87 | 2.8 |
| **Walls are made up of** | | | | |
| Clay bricks | 166 | 75.1 | 2369 | 75.2 |
| Cement blocks | 49 | 22.2 | 682 | 21.6 |
| Other | 6 | 2.8 | 100 | 3.2 |
| **Roof is made up of** | | | | |
| Asbestos sheets | 136 | 61.5 | 2019 | 64.1 |
| Roofing tiles | 68 | 30.8 | 908 | 28.8 |
| Other | 17 | 7.8 | 224 | 7.1 |
| **Floor is** | | | | |

*(Continued)*

**Table 1.** (Continued)

| Characteristic | Asthmatic | | Non-asthmatic | |
|---|---|---|---|---|
| | Number | Prevalence (%) | Number | Prevalence (%) |
| Cemented | 114 | 52.1 | 1589 | 51.0 |
| Tile | 48 | 21.9 | 615 | 19.8 |
| Concrete | 38 | 17.4 | 663 | 21.3 |
| Other | 19 | 8.7 | 246 | 7.9 |

frequency of dyspnoea symptoms was unchanged from pre-pregnancy to the first trimester (Table 4).

The prevalence of wheeze among asthma patients without comorbid cardiac conditions was 67.0% (n = 148, 95%CI 60.3–73.1). There is a significant reduction of wheezing in the first (n = 102, 46.4%) and second trimesters (n = 51, 47.7%) compared to the pre-conceptional period ($p<0.01$). Interestingly, 67.9% (n = 36) of the study participants who reported an increase in the frequency of wheezing from pre-pregnancy to the first trimester had an unplanned pregnancy. Of the 73 (33.0%) participants who did not have wheezing episodes during the last three months preceding pregnancy, new-onset wheeze was reported in the first and second trimester by 8.2% (n = 6) and 16.4% (n = 12), respectively. Demographic characteristics of study participants with new onset dyspnoea and wheeze or an increase in the symptom frequency of dyspnoea and wheeze are presented in supplementary information (S1 Table). A total of 34 (33.3%) pregnant women had at least one hospital admission due to exacerbation of wheeze during the first and second trimester.The pregnant women who sought medical care for asthma at the private sector had a lower likelihood of developing new-onset wheeze in the first trimester (unadjusted OR = 0.51; 95%CI 0.44–0.59; $p = 0.03$). None of the study participants with asthma reported a past history of habitual cigarette smoking and 20.3% (n = 40) reported that at least one family member smoked cigarettes at home. Of these 40 study participants, wheezing in the last three months preceding pregnancy, first trimester and the second trimester was reported by 62.5% (n = 25, $p = 0.50$), 46.2% (n = 18, $p = 0.97$) and 40.0% (n = 10, $p = 0.39$) asthmatic pregnant women, respectively. Using firewood for cooking ($p = 0.93$, unadjusted OR 0.99; 95%CI 0.75–1.30), having an asbestos roofing ($p = 0.36$, unadjusted OR 1.14; 95%CI 0.86–1.51), having brick walls ($p = 0.84$, unadjusted OR 1.0; 95%CI 0.75–1.42) and having a cemented floor ($p = 0.88$, unadjusted OR 0.98l 95%CI 0.75–1.29) were not associated with self-reported physician-diagnosed asthma in this cohort of pregnant women from rural Sri Lanka.

**Table 2.  Prevalence and the incidence of asthma symptoms during first and second trimester in a cohort of pregnant women (N = 3374).**

| Factor | Number | Per 1000 pregnant women | 95% confidence interval | |
|---|---|---|---|---|
| | | | Upper | Lower |
| Prevalence of a history of physician-diagnosed asthma | 223 | 66.1 | 57.9 | 75.0 |
| Prevalence of asthmatics on regular treatment | 51 | 15.1 | 11.3 | 19.8 |
| Incidence of new-onset dyspnoea in the first trimester* | 16 | 4.7 | 2.7 | 7.7 |
| Incidence of new-onset dyspnoea in the second trimester* | 16 | 4.7 | 2.7 | 7.7 |
| Incidence of new-onset wheezing in first trimester* | 6 | 1.8 | 0.6 | 3.9 |
| Incidence of new-onset dyspnoea in second trimester* | 12 | 3.6 | 1.8 | 6.2 |
| Incidence of exacerbation(s) of wheeze requiring hospitalization | 34 | 10.1 | 7.0 | 14.1 |

*New onset is defined as an asthmatic patient who did not have a particular symptom during the last three months preceding the pregnancy developing the symptom.

**Table 3. Symptomatology of asthma among self-reported physician-diagnosed asthmatic pregnant women without significant comorbid cardiac conditions in 2D echocardiogram (n = 221).**

| Symptom | Last 3 months prior to pregnancy | | During first trimester | | During second trimester | |
|---|---|---|---|---|---|---|
| | Number | % | Number | % | Number | % |
| Dyspnoea | 45 | 20.4 | 51 | 23.3 | 30 | 28.3 |
| Frequency of dyspnoea | | | | | | |
| Once a month | 9 | 22.0 | 5 | 10.4 | 19 | 82.6 |
| 2–4 times a month | 11 | 26.8 | 11 | 22.9 | 4 | 17.4 |
| Several times a week | 13 | 31.7 | 15 | 31.3 | 0 | 0 |
| Daily | 8 | 19.5 | 17 | 35.4 | 0 | 0 |
| Dyspnoea after exercise | 34 | 81.0 | 35 | 72.9 | 16 | 69.6 |
| Wheeze | 148 | 67.0 | 102 | 46.4 | 51 | 47.7 |
| Frequency of wheeze | | | | | | |
| Once a month | 48 | 35.0 | 26 | 27.1 | 24 | 55.8 |
| 2–4 times a month | 39 | 28.5 | 27 | 28.1 | 6 | 14.0 |
| Several times a week | 33 | 24.1 | 28 | 29.2 | 5 | 11.6 |
| Daily | 17 | 12.4 | 15 | 15.6 | 8 | 18.6 |
| Wheeze after exercise | 92 | 64.8 | 69 | 69.7 | 35 | 72.9 |

Of the 221 pregnant asthmatics, birthweight and preterm delivery data was obtained from hospital records for 187 and 189 participants, respectively. There were 30 (16.0%) low birthweight (LBW) deliveries and 23 (11.1%) were preterm deliveries (Table 5). Increase in the frequency of symptoms from pre-pregnancy to the second trimester for wheezing (RR 2.37, 95% CI 0.84–6.66) or dyspnoea (RR 6.4, 95%CI 0.55–74.89) was not associated with LBW.

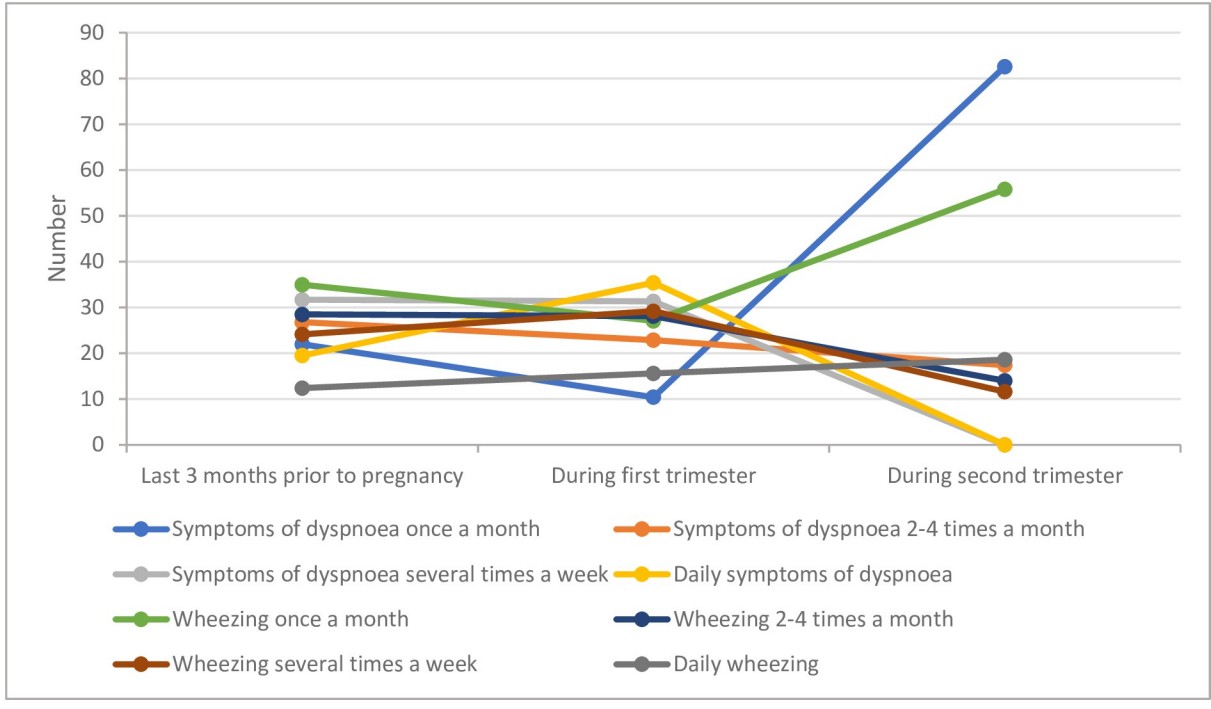

**Fig 2. Variation of the frequency dyspnoea and wheeze from the last three months before pregnancy to the second trimester among self-reported physician-diagnosed asthmatic women without significant cardiac comorbidities in 2D echocardiogram (n = 221).**

**Table 4. The change in the frequency of symptoms of wheeze and dyspnoea among pregnant women with self-reported physician-diagnosed asthma without significant comorbid cardiac conditions in 2D echocardiogram (n = 221).**

| Time period | Increased | | Unchanged | | Decreased | |
|---|---|---|---|---|---|---|
| | n | % | n | % | n | % |
| **Pre-conceptional to first trimester** | | | | | | |
| Wheeze | 53 | 25.4 | 147 | 70.3 | 9 | 4.3 |
| Dyspnoea | 1 | 3.1 | 30 | 93.8 | 1 | 3.1 |
| **First trimester to second trimester** | | | | | | |
| Wheeze | 22 | 23.4 | 46 | 48.9 | 26 | 27.7 |
| Dyspnoea | 9 | 47.4 | 2 | 10.5 | 8 | 42.1 |
| **Pre-conceptional to second trimester** | | | | | | |
| Wheeze | 38 | 40.9 | 34 | 36.6 | 21 | 22.6 |
| Dyspnoea | 9 | 50.0 | 3 | 16.7 | 6 | 33.3 |

Increased frequency of wheezing in the same period was not associated with preterm delivery as well (RR 2.32, 95%CI 0.61–8.90). Sex of the foetus was not associated with the increasing frequency of symptoms. The prevalence of anaemia in the first trimester among pregnant asthmatic women was 14.5% (n = 32) with the range of haemoglobin from 8.7 to 14.9 g/ dL. Comorbid anaemia in first trimester among pregnant asthmatic women did not further increase the risk of low birthweight (RR 0.90; 95%CI 0.29–2.8) or preterm delivery (RR 0.26; 95%CI 0.03–2.0) in this cohort.

## Discussion

The prevalence of asthma in pregnancy reported in our study (6.6%, 95%CI 5.8–7.5) is within the current reported global mean prevalence of 4–12%, though wide variations are reported in large national and multicentre studies conducted in other countries [14,15]. To the best of authors knowledge, we report for the first time the prevalence of asthma among pregnant Sri Lankan women.

In this cohort, a third of women who had an exacerbation were hospitalized: a number much larger than the estimated 20% that require interventions with only 6% requiring hospitalization [16]. Furthermore, among pregnant asthmatic women, approximately one in four reported an increase in the frequency of wheezing from pre-pregnancy to the first trimester and from the first trimester to the second trimester. More than one in ten pregnant asthmatic women (11.7%) developed new onset dyspnoea during the second trimester. These findings could be proxy indicators of poorly managed asthma during pregnancy in this study population. Furthermore, those who were under care at the private sector had a lower likelihood of developing new-onset wheeze, most probably due to the higher likelihood of specialist follow-up at private sector which is associated with better control of asthma symptoms [17].

Current evidence suggest that among asthmatic pregnant women, almost a third each had worsened, unchanged or improved asthmatic symptoms during pregnancy a finding which is almost similar to our study [18]. However, the significant reduction in the prevalence of wheeze from pre-pregnancy to the first and second trimesters could be because the majority (58.4%) of asthmatic pregnant women were not on medical treatment prior to getting pregnant.

Using firewood for cooking, which is associated with asthma [19,20], is commonly practiced in rural Sri Lanka: almost half of the households in this cohort use firewood as the major fuel source (49.7%). However, in the current study a significant difference in asthma among

**Table 5. Variation in the prevalence of low birthweight and preterm delivery among pregnant women with self-reported physician-diagnosed asthma depending on the change in symptomatology (n = 221).**

| Change in symptomatology | Low birthweight | | Preterm delivery | | Relative risk | 95% confidence interval |
|---|---|---|---|---|---|---|
| | n | % | n | % | | |
| **Increase in the frequency wheezing of** | | | | | | |
| from pre-pregnancy to first trimester | 9 | 20.5 | 5 | 11.1 | 1.05 | 0.36–3.11 |
| from pre-pregnancy to second trimester | 11 | 30.6 | 6 | 16.2 | 2.32 | 0.61–8.90 |
| from first trimester to second trimester | 6 | 27.3 | 2 | 9.1 | 0.75 | 0.15–3.83 |
| **Increase in the frequency of dyspnoea** | | | | | | |
| from pre-pregnancy to first trimester | 0 | 0 | 0 | 0 | 0.92 | 0.82–1.03 |
| from pre-pregnancy to second trimester | 4 | 44.4 | 2 | 22.2 | 2.29 | 0.17–30.96 |
| from first trimester to second trimester | 2 | 22.2 | 1 | 11.1 | 1.00 | 0.53–18.91 |

households using firewood as the major fuel source and the households using other sources of fuel—mostly liquid petroleum gas (47.5%)–was not observed. This observation may be due to firewood commonly been as a secondary fuel source, which was not assessed in this study, especially for boiling water and cooking [21]. Similarly, having an asbestos roofing is known to be associated with asthma [22]. However, in the current study the housing conditions such as the materials used for roofing, walls and the floor, did not associate with asthma or increased symptom frequency among known asthmatics. This finding highlights the crucial need to conduct further investigations to clearly understand the causative and contributory factors to bronchial asthma among pregnant women from rural geographies.

According to the national statistics of the Family Health Bureau of Sri Lanka, the prevalence of low birth-weight in Sri Lanka for the year 2019 was 12.3%, which is the highest recorded percentage since 2015 [23]. Our study shows that the prevalence of low birth-weight (16.0%) is higher in pregnant women with asthma compared to the national level prevalence which is consistent with current medical literature [24]. This may be attributable to the reduced foetal growth resulting of the hypoxia caused by chronic maternal suboptimal pulmonary function and inflammation associated with asthma [25]. Although not observed in this study sample, current evidence suggests that the incidence of preterm delivery is significantly higher among asthmatic women with inadequate symptom control compared to asthmatic women who had adequate symptom control during pregnancy [26].

Available evidence suggests that asthma in pregnancy could lead to increased risk of low birth-weight, small for gestation age, preterm delivery, congenital malformations, neonatal hospitalization and neonatal death for babies of asthmatic pregnant women [9,24]. This increased risk was associated with the severity of asthma and frequent asthma exacerbations [27]. The wide confidence intervals across the null value in our study indicates that a larger sample is required to assess the effect of asthma on these selected pregnancy outcomes. Although, anaemia is independently associated with increased risk of low birthweight [28] and preterm delivery [29], comorbid anaemia did not further increase the risk of low birthweight and preterm delivery in this cohort.

Pregnancy-related complications of asthma are dependent on the changing asthma symptom severity during pregnancy and maybe mitigated by tailored treatment aimed at asthma symptom management during pregnancy [30]. Women with controlled asthma during pregnancy has normal placental function comparable to non-asthmatic women indicating that well-controlled maternal asthma is crucial in improving the health outcomes of the children of asthmatic women [31]. Our study clearly indicates that symptoms of asthma is not considered as an important health condition by the pregnant women living in this rural community and

also the management of asthma during pregnancy is not optimal as showed by the number of hospital admissions. More focus on medical conditions complicating pregnancy is required in countries like Sri Lanka where the maternal mortality and morbidities are increasingly common due to these conditions.

## Limitations

The asthma diagnosis was based on self-reporting and only 22.6% of participants had documented evidence of asthma. Self-reported physician-diagnosed asthma is a lower level of evidence compared to asthma diagnosed with objective lung function assessment. Although 41.6% of the asthmatics were on regular medical treatment, adequate documented evidence of asthma medication of individuals was not properly available. Therefore, the effect of asthma medication on symptomatology could not be assessed in this study. We assessed the presence of cigarette smoker(s) in the household, however, further detailed examination of the degree of passive cigarette smoke exposure was not carried out. Similarly, the level of exposure to domestic risk factors were not assessed in this study. Furthermore, ideally, multivariable analysis and determination of the adjusted odds ratio for each factor should be conducted to assess the association with self-reported physician-diagnosed asthma. However, due to the small number of pregnant asthmatic women the authors did not conduct a regression analysis. In the current study, due to the disruptions caused by island-wide lockdown procedures and other COVID 19 pandemic prevention regulations, data collection in the third trimester was disrupted. Therefore, the authors do not have data on pregnancy induced hypertension, pre-eclampsia, eclampsia, gestational diabetes mellitus and other comorbid conditions that may contribute to low birthweight or prematurity in the offspring.

## Conclusion

This study reports the high prevalence of asthma and asthma-associated pregnancy outcomes in women from a rural geography signifying the importance of targeted management of asthma. Prioritizing asthma management at pre-conceptional period and early identification and management of asthma during the pregnancy is required to minimize the high rate of hospital admissions and exacerbations observed in this rural cohort of pregnant asthmatic women.

## Supporting information

**S1 Table. Demographic characteristics of patients who reported increased frequency or new onset dyspnoea or wheeze in the first and second trimester.** *New onset is defined as an asthmatic patient who did not have a particular symptom during the last three months preceding the pregnancy developing the symptom.
(DOCX)

## Acknowledgments

The authors wish to acknowledge the regional epidemiologist of Anuradhapura and the medical officers of health of Anuradhapura District for the administrative assistance, and the medical undergraduates of the Rajarata University of Sri Lanka for field support.

## Author Contributions

**Conceptualization:** Shashanka Rajapakse, Suneth B. Agampodi.

**Data curation:** Suneth B. Agampodi.

**Formal analysis:** Shashanka Rajapakse, Janith Warnasekara.

**Funding acquisition:** Thilini Chanchala Agampodi, Suneth B. Agampodi.

**Investigation:** Janith Warnasekara, Parami Abeyrathna, Gayani Amarasinghe, Ayesh Umeshana Hettiarachchi, Imasha Upulini Jayasinghe, Iresha Koralegedara.

**Methodology:** Shashanka Rajapakse, Nuwan Wickramasinghe, Janith Warnasekara, Parami Abeyrathna, Gayani Amarasinghe, Ayesh Umeshana Hettiarachchi, Imasha Upulini Jayasinghe, Iresha Koralegedara, Thilini Chanchala Agampodi.

**Project administration:** Nuwan Wickramasinghe, Ayesh Umeshana Hettiarachchi, Thilini Chanchala Agampodi, Suneth B. Agampodi.

**Supervision:** Nuwan Wickramasinghe, Thilini Chanchala Agampodi, Suneth B. Agampodi.

**Writing – original draft:** Shashanka Rajapakse.

**Writing – review & editing:** Shashanka Rajapakse, Nuwan Wickramasinghe, Janith Warnasekara, Parami Abeyrathna, Gayani Amarasinghe, Ayesh Umeshana Hettiarachchi, Imasha Upulini Jayasinghe, Iresha Koralegedara, Thilini Chanchala Agampodi, Suneth B. Agampodi.

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
