## [Decision Letter · Decision Letter 0]

10 May 2022

PONE-D-21-40055Asthma in a prospective cohort of rural pregnant women from Sri Lanka: need for better care during the pre-conceptional and antenatal periodPLOS ONE

Dear Dr. Agampodi,

Thank you for submitting your manuscript to PLOS ONE. After careful consideration, we feel that it has merit but does not fully meet PLOS ONE’s publication criteria as it currently stands. Therefore, we invite you to submit a revised version of the manuscript that addresses the points raised during the review process.

We look forward to receiving your revised manuscript.

Kind regards,

Aleksandra Barac

Academic Editor

PLOS ONE

Journal Requirements:

This research was supported by the Accelerating Higher Education Expansion and Development (AHEAD) Operation of the Ministry of Higher Education, Sri Lanka funded by the World Bank. The funding agency has no role in the design of the study and collection, analysis, and interpretation of data and in writing the manuscript. The authors wish to acknowledge the funding agency (grant identifier: DOR STEM HEMS [6026-LK/8743-LK]), regional epidemiologist of Anuradhapura and the medical officers of health of Anuradhapura District for the administrative assistance, and the medical undergraduates of the Rajarata University of Sri Lanka for field support.

Authors SA and TA received funding for this research project. This research was supported by the Accelerating Higher Education Expansion and Development (AHEAD) Operation of the Ministry of Higher Education, Sri Lanka funded by the World Bank. The funding agency had no role in the design of the study and collection, analysis, and interpretation of data and in writing the manuscript. The authors wish to acknowledge the funding agency (grant identifier: DOR STEM HEMS [6026-LK/8743-LK]).

Reviewers' comments:

Reviewer's Responses to Questions

**Comments to the Author**

1. Is the manuscript technically sound, and do the data support the conclusions?

Reviewer #1: Partly

Reviewer #2: Yes

Reviewer #3: Yes

Reviewer #4: Partly

2. Has the statistical analysis been performed appropriately and rigorously? 

Reviewer #1: No

Reviewer #2: Yes

Reviewer #3: Yes

Reviewer #4: Yes

3. Have the authors made all data underlying the findings in their manuscript fully available?

Reviewer #1: Yes

Reviewer #2: Yes

Reviewer #3: Yes

Reviewer #4: Yes

4. Is the manuscript presented in an intelligible fashion and written in standard English?

Reviewer #1: Yes

Reviewer #2: Yes

Reviewer #3: Yes

Reviewer #4: Yes

5. Review Comments to the Author

Reviewer #1: In this article, the authors evaluate a prospective cohort of pregnant women for the incidence and prevalence of asthma in a rural population in the Anuradhapura district of Sri Lanka. The work is generally technically sound. My comments are as follows.

Introduction

The physiologic changes you describe during pregnancy (decreased TLC, decreased FRC, increased minute ventilation and Vt) are generally associated with restrictive lung disease, and yet you suggest (in line 102) that these factors may be implicated in impacting asthma control. Instead, the sentences that follow (lines 102-108) are those that are implicated in asthma associated with pregnancy. Therefore, I would suggest that sentence should be edited for clarity.

Methods

It would be useful to clarify the reason patients with co-morbid significant 2D echo findings were excluded and/or define which findings would or would not prompt exclusion.

Do you mean to say (in line) that you performed multivariate analysis? If not, why did you only include bivariate analysis?

Can you clarify why RR and OR were both calculated for pregnancy outcomes?

Results

In table 2, you have two rows listing the factor “Incidence of new-onset dsypnoea in the in the second trimester”. Please clarify

In tables 3-4, can you demonstrate some statistical measure of the change in symptoms over time from pre-pregnancy to first trimester to second trimester (? Regression or Mann-Kendall)

Could table 5 go in an appendix?

In table 6, please represent the RR and Cis in the table.

Discussion

Some women with asthma improve clinically during pregnancy. Did you see this? If so, could you also address that observation (i.e. history of asthma with symptoms in the 3 months prior to pregnancy that improved in the 1st/2nd trimester).

Reviewer #2: There is no novelty in this manuscript, it just describe the situation pertaining to the asthma and pregnancy, not even followed up to the third trimester.

Diagnosis of asthma through the questionnaires without documentary evidence (only 22.9%) would have been introduced selection bias to the study. Author should pay a much attention when labelling asthmatic and non-asthmatic. Dyspnea due to other reason, need to taken into the consideration: heart failure, COPD, pneumonia or psychologic problem.

Other risk factors for asthma were not investigated: pollen, pollutants, smoking,

Need details of passive smoking

Table 2 : raw number 5 and 7 are similar, raw number 7 would be "wheezing"

How examiners were calibrated, trained were not described.

Inter examiner variability of diagnosis with clinical examination

I am not sure that all pregnant mothers were included for the study or what were the inclusion and exclusion criteria.

To complete the study all pregnant mothers should have been followed up to the termination of pregnancy.

Reviewer #3: The authors have conducted a meticulous study and reports significant results. A few corrections and clarifications are mentioned.

In the abstract and main text, it is best to stick to a similar format when reporting both numbers and percentages. If the number is within brackets such as in line 82 maintain a similar format throughout (line 171, 185 in results section also uses 2 different formats in the same section).

In the abstract under methods, line 76, the word ‘study’ should be included after ‘prospective cohort’.

It is not clear whether new onset wheeze was reported in 6 out of what population in the abstract (mentioned in the text as 73).

It is good practice to mention the lost to follow up patients in the beginning of the results section and describe the final study population, mentioning Figure 01. In line 226 it is mentioned ‘out of 221’ the proportion of LBW and preterm deliveries. However, Figure 01 states that 114 and 32 patients were lost to follow up. If so, how were the birth outcomes of all 221 assessed?

The prevalence of asthma was mentioned as 223 in line 171 and Table 02 but in Table 01 and Table 03 it is 221. In line 226, also it is mentioned as 221 pregnant asthmatics. As two were excluded due to cardiac conditions it is clearer to mention so in the beginning of the results section.

Did 53 asthma patients not seek treatment (223-84-86)? If so, how were they diagnosed? (in methods section it is mentioned that physician reported, use of inhalers, nebulization, hospitalizations and exacerbating factors were used and table 02 indicates all 223 were physician diagnosed asthma)

Table 02 and Table 06 also needs to be cited in text.

Use either simple ‘p’ or ‘P’ throughout the text for better uniformity when reporting p value.

Adjust the sentence in line 246 to better reflect the meaning.

In line 267, mention that asbestos is reported as having an association or known to. Currently it conveys that your study shows that there’s an association which is then contraindicated by the next statement.

Reviewer #4: It is a comprehensive and descriptive study, helping with having a picture of basic pregnancy care and infering problems that could be addressed in the future.

Problems with sample size and data completeness. Shaky asthma diagnosis. Good starting point for future studies but with problems in their methodology making assumptions difficult.

6. PLOS authors have the option to publish the peer review history of their article (what does this mean?). If published, this will include your full peer review and any attached files.

Reviewer #1: No

Reviewer #2: No

Reviewer #3: No

Reviewer #4: No

---

## [Author Response · Author response to Decision Letter 0]

24 May 2022

Journal requirements

Comment: Thank you for your information. We note your study included pregnant participants under the age of 18. Please ensure you have stated whether you obtained consent from parents or guardians of the minors included in the study or whether the research ethics committee or IRB specifically waived the need for their consent.

Response: We have included the following statement on obtaining assent and parental/guardian’s consent for participants under the age of 18.

“Assent and informed consent from the accompanying parent/ guardian were obtained for the research participants who are aged less than 18 years.”

Comment: Please ensure that your manuscript meets PLOS ONE's style requirements, including those for file naming.

Response: We edited the manuscript according to the PLOS ONE style requirements including titles, file names, graphs and tables.

Comment: You indicated that you had ethical approval for your study. In your Methods section, please ensure you have also stated whether you obtained consent from parents or guardians of the minors included in the study or whether the research ethics committee or IRB specifically waived the need for their consent.

Response: The methods section of the manuscript has the following statement on obtaining informed written consent.

“Informed written consent was obtained before recruiting participants to the study.” 140-141

Comment: Please note that funding information should not appear in the Acknowledgments section or other areas of your manuscript. We will only publish funding information present in the Funding Statement section of the online submission form.

Response: The information about the funding agency was removed from the acknowledgement section. 369-371

Comment: Please remove any funding-related text from the manuscript and let us know how you would like to update your Funding Statement. 

Response: Kindly keep the current funding statement.

“Authors SA and TA received funding for this research project. This research was supported by the Accelerating Higher Education Expansion and Development (AHEAD) Operation of the Ministry of Higher Education, Sri Lanka funded by the World Bank. The funding agency had no role in the design of the study and collection, analysis, and interpretation of data and in writing the manuscript. The authors wish to acknowledge the funding agency (grant identifier: DOR STEM HEMS [6026-LK/8743-LK]).” 

Comment: Please note that in order to use the direct billing option the corresponding author must be affiliated with the chosen institute. 

Response: The corresponding author Suneth Buddhika Agampodi is affiliated with the Section of Infectious Diseases, Department of Internal Medicine, School of Medicine, Yale University 43-48

Reviewer 1

The authors appreciate the constructive and insightful comments of the reviewer and has edited the manuscript accordingly. We believe that the comments improved the quality and the clarity of the manuscript.

Comment: The physiologic changes you describe during pregnancy (decreased TLC, decreased FRC, increased minute ventilation and Vt) are generally associated with restrictive lung disease, and yet you suggest (in line 102) that these factors may be implicated in impacting asthma control. Instead, the sentences that follow (lines 102-108) are those that are implicated in asthma associated with pregnancy. Therefore, I would suggest that sentence should be edited for clarity.

Response: The authors agree that this sentence should be edited to improve clarity. The sentence was edited as follows “In addition, changes in immunological functions, nutritional factors and female fetus are known to affect the control of asthma in pregnant women” 105-107

Comment: It would be useful to clarify the reason patients with co-morbid significant 2D echo findings were excluded and/or define which findings would or would not prompt exclusion.

Response: The authors agree with the reviewer and has amended the manuscript as follows “Participants (n=2) with comorbid significant 2D echo-cardiogram findings (pulmonary hypertension and ostium secundum atrial septal defect with right to left shunt) were excluded from the analysis to eliminate the role of cardiac dysfunction on the symptoms. 147-150

Comment: Do you mean to say (in line) that you performed multivariate analysis? If not, why did you only include bivariate analysis?

Response: This study was conducted as a part of a large cohort study. The study sample is the recruited study participants during data collection period.

Therefore, the number of cases in each analysis (e.g., increased dyspnoea in a given trimester) was not adequate to perform multivariate analysis.

However, the authors agree with the comment and have mentioned it as a limitation.

“Furthermore, ideally, multivariable analysis and determination of the adjusted odds ratio for each factor should be conducted to assess the association with self-reported physician-diagnosed asthma. However, due to the small number of pregnant asthmatic women the authors did not conduct a regression analysis.” 352-355

Comment: Can you clarify why RR and OR were both calculated for pregnancy outcomes?

Response: The authors thank the reviewer for the comment. This was a typing mistake and it has been corrected as follows.

“Comorbid anaemia in first trimester among pregnant asthmatic women did not further increase the risk of low birthweight (RR 0.90; 95%CI 0.29-2.8) or preterm delivery (RR 0.26; 95%CI 0.03-2.0) in this cohort.” 263-272

Comment: In table 2, you have two rows listing the factor “Incidence of new-onset dsypnoea in the in the second trimester”. Please clarify

Response: The authors agree with the reviewer and added the following footnote as requested.

* New-onset of symptoms (dyspnoea/wheeze) was defined as development of a particular symptom during the pregnancy in an individual who did not have the particular symptom last three months before pregnancy 202

Comment: In tables 3-4, can you demonstrate some statistical measure of the change in symptoms over time from pre-pregnancy to first trimester to second trimester (? Regression or Mann-Kendall)

Comment: Authors also agree with the comment. However, since we only have 3 time points (pre-pregnancy, first trimester and second trimester), we did not apply Mann-Kendall test.

Comment: Could table 5 go in an appendix?

Response: The authors moved Table 5 to supplementary information as suggested by the reviewer and renamed Table 6 as Table 5. 235

Comment: In table 6, please represent the RR and Cis in the table.

Response: The relative risk and the 95% confidence interval were provided for Table 5 (former table 6) as suggested by the reviewer. 275-278

Comment: Some women with asthma improve clinically during pregnancy. Did you see this? If so, could you also address that observation (i.e. history of asthma with symptoms in the 3 months prior to pregnancy that improved in the 1st/2nd trimester).

Response: Authors agree with the comment of the reviewer. In table 4 and Fig 2, we provided the details of the changes in the symptoms including the number and percentage of research participants who had improvement in their symptoms from pre-pregnancy to first and second trimesters. It was also discussed in the discussion section. 213-215 and 222-225

 

Reviewer 2

Comment: There is no novelty in this manuscript, it just describe the situation pertaining to the asthma and pregnancy, not even followed up to the third trimester. 

Response: This cohort study is the first reported evidence on asthma among pregnant Sri Lankan women. Furthermore, the aim of this manuscript is to describe the situation pertaining to asthma among pregnant Sri Lankan women. Authors acknowledge that follow up in the third trimester would have generated more evidence for asthma in pregnant women. However, due to the COVID 19 pandemic situation and the island-wide curfew and other restrictions imposed to curtail the pandemic, the authors were unable to collect data from research participants in the third trimester. Following relaxation of travel restrictions, authors collected information on the pregnancy outcomes.

Comment: Diagnosis of asthma through the questionnaires without documentary evidence (only 22.9%) would have been introduced selection bias to the study. Author should pay a much attention when labelling asthmatic and non-asthmatic.

Response: The authors have included the following statement on diagnosis of asthma and its limitations in the limitations section.

“The asthma diagnosis was based on self-reporting and only 22.6% of participants had documented evidence of asthma. Self-reported physician-diagnosed asthma is a lower level of evidence compared to asthma diagnosed with objective lung function assessment. Although 41.6% of the asthmatics were on current medical treatment, adequate documented evidence of asthma medication of individuals was not properly available.” 344-348

Comment: Dyspnea due to other reason, need to taken into the consideration: heart failure, COPD, pneumonia or psychologic problem.

Response: We mentioned that any participant with significant findings in 2D echocardiogram were not included in the study.

Furthermore, detailed clinical history taking and examinations were carried out by medically trained data collectors to exclude other diseases as mentioned in the methods section. 147-150 and 143-145

Comment: Other risk factors for asthma were not investigated: pollen, pollutants, smoking, Need details of passive smoking

Response: The authors have included the following statement on passive smoking in the results section.

“None of the study participants with asthma reported a past history of habitual cigarette smoking and 40 (20.3%) reported that at least one family member smoked cigarettes at home.”

Further, we have acknowledged this limitation in the limitations section.

“We assessed the presence of cigarette smoker(s) in the household, however, further detailed examination of the degree of passive cigarette smoke exposure was not carried out.” 244-246 and 349-351

Comment: Table 2: raw number 5 and 7 are similar, raw number 7 would be "wheezing" 

Response: Raw number 5 represent the participants who did not have wheeze during the last three months before pregnancy and developed wheeze in the first trimester. This includes participants who required hospital admission for symptom control and those that didn’t.

Whereas raw number 7 represent the participants who developed wheezing episodes that required hospital admission. 

Comment: How examiners were calibrated, trained were not described. Inter examiner variability of diagnosis with clinical examination

Response: Medically trained data collectors were used for data collection and clinical examination in the cohort study. “An interviewer-administered questionnaire to assess demographic factors, physical health status, symptoms of asthma and treatment seeking behaviour was used in this study after obtaining informed written consent. A positive response to the question “Have you ever been diagnosed and confirmed as having asthma?” was considered as self-reported physician-diagnosed asthma (hereafter referred to as asthma). Asthma status in participants without documented evidence was confirmed following probing for a history of inhaler use, nebulization, exacerbating factors and hospitalization.”

The detailed protocol is published as mentioned in the methods section.

“This study was conducted as part of a large prospective cohort study of pregnant women: Rajarata Pregnancy Cohort (RaPCo).[13]”

Reference: Agampodi TC, Wickramasinghe ND, Prasanna RIR, Irangani MKL, Banda JMS, Jayathilake PMB, et al. The Rajarata Pregnancy Cohort (RaPCo): study protocol. BMC Pregnancy Childbirth. 2020;20: 374. doi:10.1186/s12884-020-03056-x 152-158

Comment: I am not sure that all pregnant mothers were included for the study or what were the inclusion and exclusion criteria

Response: The authors have stated the following in the methods section.

“This study was conducted as part of a large prospective cohort study of pregnant women: Rajarata Pregnancy Cohort (RaPCo).[13] All eligible pregnant women registered with the maternal care program (almost 100% of pregnant mothers are registered with the routine program) and were residing in Anuradhapura district and having a period of gestation less than 12 weeks were invited to the study (Fig 1). The study cohort included pregnant women presented to the program from July to September 2019.”

A detailed protocol paper is already published by the authors (reference 13).

Reference: Agampodi TC, Wickramasinghe ND, Prasanna RIR, Irangani MKL, Banda JMS, Jayathilake PMB, et al. The Rajarata Pregnancy Cohort (RaPCo): study protocol. BMC Pregnancy Childbirth. 2020;20: 374. doi:10.1186/s12884-020-03056-x

133-138

 

Reviewer 3

The authors wish to appreciate the constructive and insightful comments of the reviewer. We believe that these comments improved the quality of the manuscript and we have edited the manuscript accordingly.

Comments: The authors have conducted a meticulous study and reports significant results. A few corrections and clarifications are mentioned.

In the abstract and main text, it is best to stick to a similar format when reporting both numbers and percentages. If the number is within brackets such as in line 82 maintain a similar format throughout (line171, 185 in results section also uses 2 different formats in the same section).

Response: Authors appreciate the encouraging and constructive remarks of the reviewer.

The authors agree with the reviewer about maintaining uniformity. Therefore, we edited as suggested.

“Self-reported physician-diagnosed asthma prevalence was 6.6% (n=223) with only 41.7% (n=93) on regular medical follow-up for asthma.”

“The prevalence of asthma was 6.6% (n=223; 95%CI 5.8%–7.5%) with 22.9% (n=51) participants having documented evidence of a diagnosis of asthma (Table 2).”

“Only 41.7% (n=93) of the asthmatic pregnant women reported regular medical follow-up for asthma.”

“Of the 223 pregnancies, 28.3% (n=63) were unplanned with 2.3% (n=5) due to contraceptive failure.”

“Interestingly, 67.9% (n=36) of the study participants who reported an increase in the frequency of wheezing from pre-pregnancy to the first trimester had an unplanned pregnancy.”

“by 8.2% (n=6) and 16.4% (n=12), respectively.”

“habitual cigarette smoking and 20.3% (n=40) reported that at least one family member smoked”

‘reported by 62.5% (n=25, p=0.50),”

“pregnancy, first trimester and the second trimester was reported by 62.5% (n=25, p=0.50), 46.2% (n=18, p=0.97) and 40.0% (n=10, p=0.39)”

83-85, 186-188, 189-190, 190-191, 228-229, 233, 244-245, 247 and 248

Comment: In the abstract under methods, line 76, the word ‘study’ should be included after ‘prospective cohort’.

Response: We corrected the manuscript as suggested.

“We conducted a prospective cohort study in Anuradhapura district, Sri Lanka” 78

Comment: It is not clear whether new onset wheeze was reported in 6 out of what population in the abstract (mentioned in the text as 73).

Response: We amended the sentence as suggested by the reviewer for clarity.

“Of the 73 asthmatic women who did not have wheeze in the last 3 months preceding pregnancy, new-onset wheeze was reported by 6(8.2%) and 12(16.4%) in the first and second trimester, respectively” 86-88

Comment: It is good practice to mention the lost to follow up patients in the beginning of the results section and describe the final study population, mentioning Figure 01.

Response: Authors agree with the reviewer and edited the manuscript as suggested.

“Of the 3374 participants, 221 was selected as the asthmatic cohort (2 research participants with significant findings on 2D echocardiogram were excluded). Out of the 221 research participants with asthma, mid-pregnancy assessment at 24-28 weeks is available for 107 participants. The pregnancy outcomes - birthweight and preterm delivery - were assessed using hospital birth records.” 181-186

Comment: In line 226 it is mentioned ‘out of 221’ the proportion of LBW and preterm deliveries. However, Figure 01 states that 114 and 32 patients were lost to follow up. If so, how were the birth outcomes of all 221 assessed?

Response: We included the valid percentage (LBW/189). We edited the sentence to be more meaningful as suggested.

“Of the 221 pregnant asthmatics, birthweight and preterm delivery data was obtained from hospital records for 187 and 189 participants, respectively. There were 30 (16.0%) low birthweight (LBW) deliveries and 23 (11.1%) were preterm deliveries (Table 5).” 254-256

Comment: The prevalence of asthma was mentioned as 223 in line 171 and Table 02 but in Table 01 and Table 03 it is221. In line 226, also it is mentioned as 221 pregnant asthmatics. As two were excluded due to cardiac conditions it is clearer to mention so in the beginning of the results section.

Response: The authors included the following sentence in the first paragraph of the results section as suggested.

“Of the 3374 participants, 221 was selected as the asthmatic cohort (2 research participants with significant findings on 2D echocardiogram were excluded).” 181-183

Comment: Did 53 asthma patients not seek treatment (223-84-86)? If so, how were they diagnosed? (in methods section it is mentioned that physician reported, use of inhalers, nebulization, hospitalizations and exacerbating factors were used and table 02 indicates all 223 were physician diagnosed asthma)

Response: We used self-reported physician diagnosed asthma to identify patients with asthma. During data collection, further questioning on the use of inhalers, nebulization, hospitalization and exacerbating factors were used to clarify whether the participants reporting physician diagnosed asthma actually had asthma.

Of the 223 (including the two participants with significant comorbid cardiac conditions) 93 were on regular medical follow up for asthma. Others sought medical care only if they suffer an acute asthma exacerbation.

For clarity, we changed the term “current medical follow up” to “regular medical follow up” 84, 190, 347

Table 2

Comment: Table 02 and Table 06 also needs to be cited in text.

Response: Table 2 is cited in line 182 and Table 6 (renamed as Table 5 following the suggestion of one reviewer to move former table 5 to supplementary information) is cited in line 237

188, 256

Comment: Use either simple ‘p’ or ‘P’ throughout the text for better uniformity when reporting p value.

Response: As suggested, the authors changed to “p” for uniformity throughout the text. 90, 206, 208, 229, 244

Comment: Adjust the sentence in line 246 to better reflect the meaning.

Response: The sentence was edited to better reflect the meaning as suggested by the reviewer.

“In this cohort, a third of women who had an exacerbation were hospitalized: a number much larger than the estimated 20% that require interventions with only 6% requiring hospitalization” 286-288

Comment: In line 267, mention that asbestos is reported as having an association or known to. Currently it conveys that your study shows that there’s an association which is then contraindicated by the next statement.

Response: The authors edited the sentence as suggested by the reviewer.

“Similarly, having an asbestos roofing is known to be associated with asthma” 307-308

 

Reviewer 4

The authors appreciate the encouraging and constructive comments of the reviewer. We edited the manuscript accordingly.

Comment: It is a comprehensive and descriptive study, helping with having a picture of basic pregnancy care and infering problems that could be addressed in the future.

Response: Authors appreciate the encouraging comments of the reviewer. 

Comment: Problems with sample size and data completeness. 

Response: The authors agree with the reviewer and acknowledge the impact of the COVID-19 pandemic and the restrictions imposed to curtail the pandemic limited the opportunities for data collection. 356-361

Comment: Shaky asthma diagnosis.

Response: Authors agree with the reviewer and acknowledge the limitations in using self-reported asthma. We have included this as a limitation of the study. 344-348

Comment: Good starting point for future studies but with problems in their methodology making assumptions difficult.

Response: Authors agree that further studies are needed to understand the impact of asthma on pregnant women from rural geographies.

---

## [Editor Report · Decision Letter 1]

30 May 2022

Asthma in a prospective cohort of rural pregnant women from Sri Lanka: need for better care during the pre-conceptional and antenatal period

PONE-D-21-40055R1

Dear Dr. Agampodi,

We’re pleased to inform you that your manuscript has been judged scientifically suitable for publication and will be formally accepted for publication once it meets all outstanding technical requirements.

Kind regards,

Aleksandra Barac

Academic Editor

PLOS ONE
---

## [Editor Report · Acceptance letter]

7 Jul 2022

PONE-D-21-40055R1 

Asthma in a prospective cohort of rural pregnant women from Sri Lanka: need for better care during the pre-conceptional and antenatal period 

Dear Dr. Agampodi:

I'm pleased to inform you that your manuscript has been deemed suitable for publication in PLOS ONE. Congratulations! Your manuscript is now with our production department. 

Kind regards, 

on behalf of

Dr. Aleksandra Barac 

Academic Editor

PLOS ONE